# Development of Artificial Plasma Membranes Derived Nanovesicles Suitable for Drugs Encapsulation

**DOI:** 10.3390/cells9071626

**Published:** 2020-07-06

**Authors:** Carolina Martinelli, Fabio Gabriele, Elena Dini, Francesca Carriero, Giorgia Bresciani, Bianca Slivinschi, Marco Dei Giudici, Lisa Zanoletti, Federico Manai, Mayra Paolillo, Sergio Schinelli, Alberto Azzalin, Sergio Comincini

**Affiliations:** 1Department of Biology and Biotechnology, University of Pavia, 27100 Pavia, Italy; carolina.martinelli01@universitadipavia.it (C.M.); fabio.gabriele01@universitadipavia.it (F.G.); elena.dini01@universitadipavia.it (E.D.); francesca.carriero01@universitadipavia.it (F.C.); giorgia.bresciani01@universitadipavia.it (G.B.); bianca.slivinschi@gmail.com (B.S.); marco.deigiudici01@universitadipavia.it (M.D.G.); lisa.zanoletti01@universitadipavia.it (L.Z.); federico.manai01@universitadipavia.it (F.M.); alberto.azzalin@unipv.it (A.A.); 2Department of Drug Science, University of Pavia, 27100 Pavia, Italy; mayra.paolillo@unipv.it (M.P.); sergio.schinelli@unipv.it (S.S.)

**Keywords:** extracellular vesicles, nanomedicine, drugs-delivery

## Abstract

Extracellular vesicles (EVs) are considered as promising nanoparticle theranostic tools in many pathological contexts. The increasing clinical employment of therapeutic nanoparticles is contributing to the development of a new research area related to the design of artificial EVs. To this aim, different approaches have been described to develop mimetic biologically functional nanovescicles. In this paper, we suggest a simplified procedure to generate plasma membrane-derived nanovesicles with the possibility to efficiently encapsulate different drugs during their spontaneously assembly. After physical and molecular characterization by Tunable Resistive Pulse Sensing (TRPS) technology, transmission electron microscopy, and flow cytometry, as a proof of principle, we have loaded into mimetic EVs the isoquinoline alkaloid Berberine chloride and the chemotherapy compounds Temozolomide or Givinostat. We demonstrated the fully functionality of these nanoparticles in drug encapsulation and cell delivery, showing, in particular, a similar cytotoxic effect of direct cell culture administration of the anticancer drugs. In conclusion, we have documented the possibility to easily generate scalable nanovesicles with specific therapeutic cargo modifications useful in different drug delivery contexts.

## 1. Introduction

Exosomes, microvesicles and apoptotic bodies are generally referred to as extracellular nanovesicles (EVs) with a 40–5000 nm diameter size range that are released by cells in constitutive or inducible conditions [1,2,3]. EVs have been detected in the main body fluids as peripheral blood, milk, urine, saliva, plasma, and cerebrospinal fluid and therefore can be used as liquid biopsies. EVs are composed of a membrane and cytoplasmic proteins, lipid moieties, and nucleic acids molecules. The physiological/pathological state of the cells of origin directly influence the molecular architecture and composition of EVs [4,5]. EVs can transfer proteins, lipids, and nucleic acids (DNA, mRNAs and miRNAs) in short to long intercellular ranges, facilitating intercellular communication [6], and thus producing a homotypic or heterotypic transfer of biological material [7,8]. EVs, previously considered as cellular by-products [9], are presently viewed as important inducers and regulators of physiological and pathological pathways [10]. Indeed, EVs have relevant counterparts in coagulation [11], pregnancy [12], and immune response [13] processes. Furthermore, EVs are released in large quantities in pathological contexts such as infection, inflammation [14], or cancer [15,16], regulating the disease associated processes.

The prevalent therapeutic clinical approaches for human high grade glial tumors (i.e. gliomas or astrocytomas) indicate a combination of surgery, radiation, and chemotherapy treatments. The chemotherapy approach is mainly focused on the administration of Temozolomide (TMZ), a DNA alkylating agent that induces epigenetic modifications (i.e., methylation of the O6-methylguanine–DNA methyltransferase promoter), resulting in an increase of the prognostic rates [17,18]. In addition to TMZ, Bevacizumab, a humanized monoclonal antibody approved by Food and Drug Administration (FDA) in glioma therapy, acts as an anti-angiogenic moiety that efficiently interferes with ectopic tumor vasculature by down-regulating the expression of the vascular endothelial growth factor (VEGF-A) [19]. However, these treatments, along with many others based on drugs, are not completely meaningful due to limited pharmacokinetics uptake and specificity and to possible off-target effects. Therefore, novel treatment strategies characterized by enhanced efficiency and specificity and producing less disadvantageous effects, are rapidly required [20]. To this regard, nanomedicine is becoming a promising additional discipline, contributing to increased diagnostic and therapeutic abilities. Importantly, specific EVs loaded with different therapeutic compounds could improve therapeutic efficacy and specificity, favoring the prolongation of the half-life of their circulation into body fluids, exhibiting a more precise cell-target uptake, and showing a better monitored pharmacological delivery [21]. For high grade glial tumors, particularly for glioblastoma multiforme, different EVs have been evaluated in their efficacy in early phase clinical trials, including liposomes polymers and inorganic nanoparticles [22]. It has been reported that EVs share numerous beneficial properties in the specific shuttle of therapeutic drugs, especially, in their capacity to diffuse into different tissues and cellular junctions [23]. Notably, since therapeutic EVs can originate from autologous cells, their possible adverse effects should be significantly reduced [24]. Furthermore, EVs may exhibit specific targeting features with molecular cellular or tissue affinities [25]. This feature is relevant to preferentially supply therapeutic compounds to their designed cells/tissues destinations, thus reducing off-target effects [26]. Recently, different approaches have been investigated to exploit the features of mimetic nanovesicles (M-NVs) that, unlike natural EVs derived from cellular physiological processes, are generated in vitro via assembling different sub-cellular components [27,28,29]. These de novo vesicles resemble biophysical and biochemical features of nanovesicles and can be produced in larger quantities, thus facilitating their use in therapeutic protocols. To this end, recently, a systematic classification for M-NVs has been suggested [30]. Based on this nomenclature, artificial or mimetic EVs can be grouped into semi-synthetic and synthetic categories. Semi-synthetic M-NVs include engineered EVs obtained by means of genetic or metabolic handling of parental growing cells or following EVs post isolation processes. On the other hand, synthetic EVs are further partitioned into two subgroups, respectively termed “bottom top”, or “top down”: in the former, EVs are produced from distinct molecular components (i.e., proteins, nucleic acids and lipids), while for the latter, nanovesicles are generated from different portions of parental cells, mostly represented by plasma membranes constituents [30].

In the present contribution, we describe a simplified “top down” approach to generate M-NVs with the possibility to efficiently modify their intravesicular cargo with therapeutic drugs or molecules.

## 2. Materials and Methods

### 2.1. Isolation and Characterization of Extracellular Vesicles (EVs)

T98G (#CRL-1690) and U138-MG (#HTB-16) established human malignant glioma (ATCC, Guernsey, Ireland) and Res259 low grade astrocytoma (described in [31]) cell lines were cultivated in D-MEM medium supplemented with 10% FBS, 100 units/mL penicillin, 0.1 mg/mL streptomycin and 1% L-glutamine (Euroclone, Milan, Italy), at 37 °C and 5% CO_2_ atmosphere. For isolation of EVs, cells were grown in 75 cm^2^ flasks at 75% confluence using Exosome-depleted FBS (Thermofisher, Waltham, MA, USA), provided 24 h before EVs isolation.

The extraction and purification of EVs from cell culture media were conducted as described [32], by differential centrifugation and filtration. In particular, centrifugation at 400× *g* for 10 min was performed to sediment a main portion of the cells, at 2000× *g* to remove cell debris, and at 10,000× *g* to remove the aggregates of biopolymers, apoptotic bodies, and the other structures with the buoyant density higher than that of EVs. EVs contained in the resulting supernatant were sedimented by ultracentrifugation at 150,000× *g* for 2  h. The non-EV proteins in the pellet were removed by suspension followed by repeated ultracentrifugation. The obtained EVs were further purified according to their size by microfiltration filters of 0.45 μm pore diameter, as recommended [32,33].

Exosomes were isolated using the “total exosome isolation (from cell culture media) reagent” (Invitrogen, Carlsbad, CA, USA) as described [34]. In detail, culture media were collected, and 0.5 volumes of total exosome isolation reagent was added to the media and incubated at 4 °C for 20 h. Then, exosomes were pelleted at 12,000× *g* for 60 min at 4 °C and finally resuspended into 500 μL of sterile ice-cold D-PBS (Thermofisher) and next filtered (0.20 μm). 

The qNano Gold instrument (Izon Science, Christchurch, New Zeeland) was employed to measure the size distribution, concentration and z-charge of the isolated nanoparticles (EVs, exosomes and M-NVs) using the Tunable Resistive Pulse Sensing (TRPS) principle [35]. Briefly, 35 μL of purified particles were analyzed with qNano Gold instrument using a NP200 Nanopore (Izon Science) and applying 46 mm stretch, 0.34 V, and 8 mBar parametric conditions as described [36]. The calibration particles (CPC100, Izon Science) were assayed before the experimental samples under identical conditions. Size, concentrations (2000 events each) and z-charge (500 events) values of particles were finally determined using the qNano software provided by Izon Science (Izon Control Suite version 3.1).

Nanoparticles were then visualized by transmission electron microscopy (TEM). In detail, 20 μL drops of the isolated particles in D-PBS were placed on a Parafilm (Sigma, St. Louis, MI, USA) sheet, and a 300-mesh nickel grid (covered with a Formvar-carbon film) was floated onto the drops and allowed to stay for 5 min. The grids were rapidly blotted with filter paper and negatively stained with a 2% phosphotungstic acid solution, pH 7.0, for 60 s, blotted on paper and observed directly on a Zeiss EM900 electron microscope (Zeiss, Oberkochen, Germany) operating at 80 kV.

### 2.2. Generation of M-NVs

Mimetic nanovesicles (M-NVs) were obtained after trypsinization and lysis of about 5 × 10^6^ T98G cells as follows. Cell pellets were incubated for 60 min on ice in 500 μL of RIPA buffer (radioimmunoprecipitation assay buffer) (150 mM sodium chloride, 1.0% NP-40, 0.5% sodium deoxycholate, 0.1% SDS, 50 mM Tris, pH 8.0), widely adopted for whole cell lysis. Lysates were then centrifuged at 4 °C for 20 min at 12,000 rpm. Supernatants were then removed, while pellets from membranes debris were washed twice with ice cold D-PBS and finally resuspended into 500 μL D-PBS. M-NVs were then further purified by microfiltration filters with 0.45 μm pores diameter. To spontaneously generate vesicles, fragmented filtered membranes were then incubated at 37 °C for 1 h, quantified using qNano Gold instrument, and visualized by TEM as reported above. 

### 2.3. CD63 and PHK Membranes Staining of M-NVs or Exosomes

An anti-CD63 mouse monoclonal antibody, Alexa Fluor 633 conjugated (#H5C6, Biolegend, San Diego, CA, USA) and the lipophilic dyes PHK26 or PHK67 (Merck, Darmstadt, Germany) were used to stain the membranes of M-NVs or exosomes. A goat anti-rabbit Alexa Fluor 633 conjugated secondary antibody (0.1 μg) (#A-21070, Thermofisher) was used to assay its intravesicular incorporation into nascent M-NVs (~2 × 10^7^ particles). CD63 antibody (0.1 μg) was centrifuged for 10 min at 17,000× *g* before use to eliminate aggregates and then incubated for 1 h at room temperature with M-NVs or exosomes (both ~2 × 10^7^ particles) with 1% BSA (*v*/*v*). Then, stained particles were purified from unstained dyes using exosome spin columns (MW 3000, Thermofisher) as recommended. M-NVs or exosomes (both ~2 × 10^7^ particles) were stained for 5 min at room temperature in agitation with PHK26 or PHK67 dyes (0.5 μL each in a final volume of 100 μL using diluent buffer C) and columns purified as described above. 

### 2.4. pEGFP Electroporation of M-NVs

pEGFP expression vector (0.5 μg), described in [37], was electroporated into 100 μL of purified M-NVs particles (~2 × 10^7^) resuspended into D-PBS, using the NEON Transfection System (Thermofisher), at ΔV = 600 V, pulse width = 10 ms, pulse number = 20. Electroporated M-MVs were directly administered to ~2 × 10^3^ T98G cells, grown on a coverglass and fixed after 72 h p.t as described [38]. Then, cells were treated with anti-fade with DAPI reagent (Thermofisher) and visualized by fluorescence microscopy using a Nikon Eclipse TS100 (100×).

### 2.5. Berberine, Temozolomide or Givinostat In Vitro Treatments

Berberine chloride and Temozolomide (TMZ), both from Merck, and Givinostat, GVS, (described in [39]) were resuspended into DMSO (Merck) at a final concentration of 5 mM. To load these compounds separately into M-NVs, freshly isolated membranes after RIPA buffer lysis, centrifugation and filtration, isolated from a nearly confluent 75 cm^2^ flask of T98G cells (~5 × 10^6^) were quantified by qNano Gold instrument (~2 × 10^10^) and incubated at 37 °C for 1 h in the presence of Berberine, TMZ, or GVS to a final concentration of 50 μM each. M-NVs were then columns purified as above described. In parallel, Berberine, TMZ or GVS (all at 50 μM) were directly administered to growing T98G cells.

### 2.6. Viability and Apoptosis Cytofluorimetric Analysis

The viability and apoptotic rates of glioma cells were quantified using the Muse Count & Viability or Annexin V and Dead Cell assays (Luminex, Austin, TX, USA), as described [40]. After trypsinization and collection, cells were washed 3 times with D-PBS, resuspended in D-PBS + 1% FBS (*v*/*v*) and 1 volume each of Count & Viability or Annexin V reagents, and incubated for 5 or 20 min at room temperature in the dark, respectively. The analysis were then performed using the Muse cell flow cytometer analyzer (Luminex).

### 2.7. Imagestream Analysis

EVs or cells samples were analysed using the ImageStreamX MarkII instrument (ISX; Amnis/Luminex) equipped with 3 lasers (100 mW 488 nm, 150 mW 642 nm, 70 mW 785 nm (SSC). The ISX objectives, 40× (NA = 0.75; DOF = 4 µm) and 60× (NA = 0.9; DOF = 2.5 µm) were adopted. All lasers were set to maximum powers; cells data were acquired using a 40× magnification (core size = 10 µm), while EVs with 60× magnification (core size = 7 µm). PHK67 and Berberine signals were collected in channel 2 (480–560 nm filter) while anti-CD63 Alexa Fluor 633 conjugated primary antibody, anti-goat Alexa Fluor 633 secondary antibody and PHK26 dye using channel 5 (595–650 nm filter). Channel 6 (745–800 nm filter) was used for scatterplot (SSC) detection. Standard sheath fluid (D-PBS, Themofisher) without further filtration was used in all measurements. Negative controls for M-NVs and exosomes included detergent lysis controls, buffer controls without particles and unstained antibody samples. Data analysis was performed using Amnis IDEAS software (version 6.1). The gating strategies used are described in the Results section and Figure legends.

### 2.8. Statistical Analysis

The data were analyzed using the statistics functions of the MedCalc statistical software version 18.11.6. (http://www.medcalc.org). The ANOVA test differences were considered statistically significant when *p* ≤ 0.05.

## 3. Results

Nearly confluent (~75%) T98G glioblastoma cells, originally splitted into two 75-cm^2^ flasks, were separately subjected to EVs isolation by ultracentifugation, and to plasma membranes isolation RIPA (described in Section 2.1 and Section 2.2, respectively). Ultracentrifugation isolated EVs were resuspended into 500 µL of D-PBS and kept on ice. Plasma membranes mimetic nanovescicles (M-NVs) obtained after RIPA buffer incubation and subsequent centrifugation were purified using a 0.45 µm filter and incubated for 1 h at 37 °C. The two preparations (i.e., EVs by ultracentrifugation and M-NVs through plasma membranes reassembling) were then compared in nanoparticles diameter size and concentration by means of TRPS analysis using the qNano Gold instrument. As reported in Figure 1, the two approaches produced nanoparticles with similar diameters (as defined by minimum, maximum, mean and mode values) and concentration (1.99 × 10^10^ and 2.37 × 10^10^ particles/mL, respectively for ultracentrifugation isolated EVs and M-NVs).

To further characterize the morphological and biochemical features of M-NVs, they were compared to exosomes isolated from the same growing T98G cells. Differences in particles concentrations (M-NVs = 2.7 × 10^10^ particles/mL; exosomes = 6.8 × 10^11^ particles/mL) as well as in size distribution are reported, while z-charge values were roughly similar (M-NVs: *z* = −21.4 mV; exosomes: *z* = −20.7 mV) (Figure 2). Notably, TEM analysis highlighted for M-NVs a reduction of detectable intravesicular content, in contrast to exosomes particles.

To confirm ultrastructure intravesicular differences, total RNA was extracted and quantified from M-NVs and exosomes, both consisting of 2 × 10^10^ particles, derived from T98G cells. Fluorimetric analysis by Qubit RNA high sensitive kit (Thermofisher) revealed a low out of range signal for M-NVs, while 2 ng of total RNA was obtained from the exosomes sample.

To characterize some molecular features of the investigated particles, exosomes and membrane-generated particles isolated from T98G cells, after quantification by qNano Gold instrument (~2 × 10^7^ total particles) and purification, were separately incubated with an anti-CD63 Alexa Fluor 633 conjugated antibody. Particles were then further columns purified (as described in 2.3 of the Methods) and analysed by ImageStreamX MarkII flow cytometry. As reported in Figure 3, CD63 + most enriched fractions were R1 (23.5%) and R2 (45.1%) gates, respectively, for exosomes and M-NVs.

To demonstrate the biological functionality of the membranes-generated particles, i.e., the possibility to be efficiently internalized into growing cells, plasma membranes isolated from T98G cells, purified as described in the Methods, were stained with the PHK26 dye. After incubation, red fluorescent generated M-NVs were columns purified and analysed by ImageStreamX MarkII flow cytometry. As documented in Figure 4A, the scatterplot intensity graph highlighted again two main particles size-related sub-populations (i.e., R3 and R4), likely associated with exosomes and higher nanoscales M-NVs, respectively. Microscope associated analysis (60× magnification) did not reveal bright-field signals, as expected; differently, R3 and R4 representative particles showed associated red fluorescent signals (Figure 4A, right panels). Next, PHK26-stained and unstained M-NVs were separately administered to T98G growing cells; after overnight incubation, cells were fixed, DAPI stained and visualized using fluorescence microscopy (100× magnification). As reported, cytoplasmic fluorescent dots were largely evidenced compared to cells treated with PHK26 unstained M-NVs (Figure 4B).

To exclude the detected spots derived from unspecific fluorescence, M-NVs (2 × 10^7^ particles) were isolated as described from T98G RIPA-lysed cells and incubated for 1 h at 37 °C with PHK26 dye (0.5 µL) and finally purified using Exosome Spin columns. In parallel, to test the efficiency of the chromatographic columns to remove unincorporated dyes, the same amount of PHK26 dye was firstly introduces into a column. Then, the eluted solution was incubated with M-NVs (2 × 10^7^ particles). Finally, the samples were analysed in flow cytometry (Amnis ImageStreamX MarkII). As reported in Appendix A, unspecific fluorescent signals were largely reduced by column purification compared to those derived from M-NVs staining and purification (0.3 vs. 45.6%, respectively). In addition, no intracellular fluorescent evidences were detected after the administration of PHK26 column-purified sample compared to M-NVs stained one.

To further characterize the kinetics of M-NVs internalization, similarly as above, membranes isolated from T98G cells, stained with the PHK67 green fluorescent plasma membranes linker, after removal of excessive misincorporated dyes by size exclusion columns, were administered as ~2 × 10^7^ qNano Gold quantified particles to growing T98G cells and visualized at different time intervals (i.e., 0.5–1–2–24 h) using inverted fluorescence microscope (40× magnification). Fluorescent M-NVs were timely and progressively internalized, showing a massive signal evidence after overnight incubation (Figure 5). 

To further elucidate into one of the main biophysical feature of extracellular vesicles, i.e., the possibility to encapsulate molecules and transfer their modified cargo into recipient cells, membranes-generated M-NVs from T98G cells, were electroporated with an eukaryotic vector expressing the exogenous green fluorescent protein (i.e., pEGFP). Then, electroporated M-NVs were column purified and administered to growing T98G cells and, after 72 h incubation, fixed, DAPI-stained, and visualized by fluorescence microscopy at 100× magnification. As illustrated, discrete green cytoplasmic fluorescent signals were detected (Figure 6A). Flow cytometric analysis of T98G cells after overnight administration of M-NVs electroporated with pEGFP estimated that 94.5% of the analysed cells displayed fluorescent spots (Figure 6B).

To further consider the possibility to spontaneously encapsulate molecules into membranes-generated nanovesicles, ~2 × 10^7^ T98G-isolated M-NVs were stained, as before, with PHK67 green dye (0.5 µL) and incubated with of anti-goat Alexa Fluor 633 conjugated secondary antibody (0.1 µg). M-NVs were then columns purified from excessive fluorescent dyes and visualized in flow cytometry (Amnis ImageStreamX MarkII). As reported, particles of R3 and R4 scatterplot populations exhibited 51.7% of fluorescent co-localized yellow signals (Ch.2/Ch.5) (Figure 7).

Next, to evaluate the cargo capabilities of the generated M-NVs in hosting potentially therapeutics drugs, T98G mimetic M-NVs were incubated with the autoflorescent plant alkaloid Berberine. To this purpose, M-NVs (~2 × 10^10^) isolated from T98G cells (~5 × 10^6^) were incubated with 50 µM Berberine at 37 °C for 1 h, columns purified as described and administered at a final concentration of 10 µM to T98G cells, growing at a 75% confluence in a 75 cm^2^ flask. After 24 h incubation, cells were trypsinized and analysed by flow cytometry. As illustrated in Figure 8, the large majority of cells exhibited diffuse fluorescent signals compared to untreated cells, as revealed by cytometry (i.e., 87.4%) and fluorescence microscope analysis.

Taking advance of the autofluorescence feature of Berberine, a fluorimetric assay was performed to measure the incorporation of the compound into T98G plasma membranes-generated M-NVs (Appendix A). Firstly, kinetics curve of the fluorescent emission of Berberine alone (i.e., 0–1–10–100–1000 µM) or in presence of M-NVs (M-NVs = 2 × 10^7^/mL, in a volume of 30 µL, thus having 6 × 10^8^ total particles, as quantified using qNano Gold instrument) was calculated. Fluorescence logarithm interpolation of M-NVs incubated with 1000 µM Berberine (i.e., logF M-NVs = 3.28) determined a fluorescent signal roughly corresponding of that produced by 10 µM Berberine measurement. Finally, stoichiometric calculations and the final ratio estimated 2 × 10^6^ encapsulated Berberine molecules per single M-NV particle.

Then, to compare the effect of direct drugs administration in culture with those mediated by drugs-loaded M-NVs, again, T98G derived plasma membranes were isolated, quantified by qNano Gold instrument (~2 × 10^10^ particles) and incubated with Temozolomide (TMZ) or Givinostat (GVS) (both at 50 µM). Subsequently, M-NVs were purified as described and administered to three different astrocytoma established cells, i.e., T98G, U138-MG, and Res259 (each at ~5 × 10^6^ cells into 75 cm^2^ flasks) The same cells were treated in parallel with a corresponding amount of the investigated drugs. After 24 h of incubation, cells were trypsinized and their viability was measured by cytofluorimetric assays (Muse, Cell Count and Viability, Merck). As highlighted in Figure 9, M-NVs loaded with TMZ/GVS induced similar decreased trends in viability compared to the direct drugs administration schemes; altogether, the three different astrocytoma-derived cells displayed similar phenotypic behavior. Similar viability trends were observed in the same experiment after 48 h of incubation (data not shown).

To deeply investigate the observed TMZ cytotoxic effect, through direct culture administration or mediated by plasma membranes-generated M-NVs loaded with the drug, T98G cells were analysed in apoptotic cell death induction. TMZ (50 µM) or M-NVs (~2 × 10^10^ particles, loaded or not- with TMZ 50 µM) were added to growing T98G cells (~5 × 10^6^). After 24 h, cells were trypsinized and analysed for Annexin V expression by flow cytometry (Annexin V, Muse, Luminex). As a result, TMZ and M-NVs-loaded with TMZ induced a marked increase in early apoptosis events, compared to mock treated cells (i.e., 42.8% and 40.4%, respectively) (Figure 10), thus producing a significant increase in apoptotic induction. Similar apoptotic profiles were observed in the same experiment after 48 h of incubation (data not shown).

## 4. Discussion

One of the main technological challenge for the clinical employment of anticancer drugs is to establish a suitable methodology for the specific targeting of the therapeutic molecules to their final cellular destination. Improvement in pharmacokinetics of these drugs are therefore long overdue, and for this reason, biophysical and biochemical properties of EVs have been deeply studied to establish novel treatment horizons [26].

Alternative strategies for EVs isolation have been reported, and focused on different features such as density, size, morphology, and on the differential expression of surface markers and receptors. To this purpose, ultracentrifugation protocols, the addition of coating precipitating polymers, size exclusion chromatography, and the employment of specific magnetic beads expressing anchor peptides have been extensively documented [41,42,43]. Since EVs are cellular released products, their number depends on the ability to maintain a large cell population in culture. A consequent drawback to increase EVs production is the needed presence of animal serum for ideal cell culturing conditions. Indeed, it was reported that fetal bovine serum has a relatively high content of parental animal-derived EVs [44], thus potentially introducing exogenous EVs impurities. On the other hand, it was reported that EVs isolated from cell culture media in the absence of serum are enriched in reactive oxygen species and stress-related proteins, reflecting stress-induced phenotypic changes as starvation in cell culture [44]. As already stated, therapeutic systems based on EVs loaded with drugs displayed crucial benefit compared to other modalities, particularly in avoiding degradation processes of the therapeutic compounds and in enhancing the ability of drugs-loaded EVs to cross many biological barriers. Of particular interest is the difficulty in delivering therapeutic compounds into the brain tissues for the treatment of central nervous system diseases such as glial tumors. In fact, it was reported that EVs have been shown to cross the blood–brain barrier through the olfactory region as a prerequisite step that culminates in their internalization into neuronal, microglial, and oligodendrocyte cells [45].

Recently, the use of EVs as therapeutic tools has been associated with the design and development of mimetic nanovesicles (M-NVs). Indeed, M-NVs require relatively simple isolation procedures with the potentiality of high yield particles formation and exhibiting similar biophysical and biochemical features of natural EVs [27]. Importantly, M-NVs can be directly loaded with different molecules (nucleic acids, peptides, drugs) during their formation processes. After the mechanical or biochemical fragmentation of the plasma membranes of the parental cells, the therapeutic compounds in the buffer can be embedded into nanovesicles by the spontaneous reassembly of the plasma lipid membranes. Notably, this M-NVs packaging process does not affect the parental cell biology [27].

In the present contribution, we described a bench-scaling up affordable method to generate mimetic drugs-loaded EVs form cell culture platforms. This proof-of-principle development is based on the cell lysis induced by ionic detergents SDS and sodium deoxycholate as active constituents (i.e., RIPA buffer), previously reported as reliable tool for the isolation of EV peptides and proteins [46]. The rationale of our proposed protocol is based on the hypothesis that not all components of native EVs are crucial for cargo delivery, in particular the intravesicular molecular content [47]. Therefore, from a structural and biochemical point of view, EVs can be considered as lipid micelles with attached proteins as suitable templates to develop mimetic EVs [47]. According to our reported results, the fragmentation of the cell plasma membranes by the incorporation of detergents into the lipids bilayers produced functional templates able to spontaneously reassembly into novel nanovesicles. This strategy relies on the principle of self-assembly of lipids and protein membranes into spherical structures with the contemporary encapsulation of the surrounding molecules into the aqueous cavity of the resulted nanovesicles [30]. Importantly, RIPA buffer was reported not sufficient to solubilize membrane proteins and membrane-associated proteins concentrated in lipid rafts. These are highly specialized microdomains on the lipid bilayers which contain specialized lipids, cholesterol and functional proteins. These lipid rafts, also referred as “detergent resistant membrane” (DRM), are usually insoluble by mild detergent buffers such as 1% Triton X-100 and RIPA buffer [48].

As a result, the spontaneously generated nanovesicles showed the expected size morphology and z-charge values and also displayed by TEM analysis a relatively empty intravesicular content, a volume suitable for hosting a multitude of exogenous molecules. Furthermore, the efficiency of cellular uptake demonstrated that the fluorescent dyes decorated mimetic vesicles were functional in their biochemical stability.

EVs have been widely reported in literature as efficient transporters of different therapeutic compounds [49]. For example, exosomes loaded with Curcumin, a component of the golden spice turmeric (*Curcuma longa*) able to modulate different cell signaling pathways, were obtained by direct mixing [50], Paclitaxel, a chemotherapic compound largely employed for the treatment of different cancers, was introduced into EVs through incubation and ultrasonic treatments [51,52]. In contrast, M-NVs produced by membrane extrusions processes efficiently incorporated the chemotherapic drug Doxorubicin [53]. In particular, Kalimuthu and collaborators [49] showed that by increasing Paclitaxel concentration, the size of produced EVs by membrane extrusion increased, thus inducing an increase in the bioavailability of the EV-incorporated drug inside the target cell. Following this indication, an excess of autofluorescent Berberine chloride, a bioactive compound extracted from different plants and reported to inhibit proliferation and promote cytotoxicity towards cancer cells [54], was incubated with plasma membrane reassembled M-NVs isolated from T98G cells and subjected to RIPA lysis. Our flow cytometry and microscope data showed that the produced M-NVs efficiently incorporated the compound, next internalized into tumor cells. Following the described protocol, the chemotherapic drug Temozolomide [17] or the pan-histone deacetylase inhibitor Givinostat [39] were separately added to the forming nanoparticles. The reliability of our approach is further supported by our finding that astrocytoma cultured cells, upon treatment with our drug-loaded M-NVs, displayed a significant decrease of cell viability and a concomitant significant increase in apoptotic rate.

In conclusion, we have described a simplified method to generate biological functional mimetic nanovesicles that might be easily applied to different plasma membrane templates with the possibility to scale up their synthetic capability. In addition, this procedure does not require any specific pressurization or sophisticated microfluidic plasma membrane extruding devices to generate mimetic nanoparticles. Furthermore, the spontaneous encapsulation process might be optimized for specific pathological treatments, even considering different drug combinations to increase the therapeutic efficacy.

## Figures and Tables

**Figure 1 cells-09-01626-f001:**
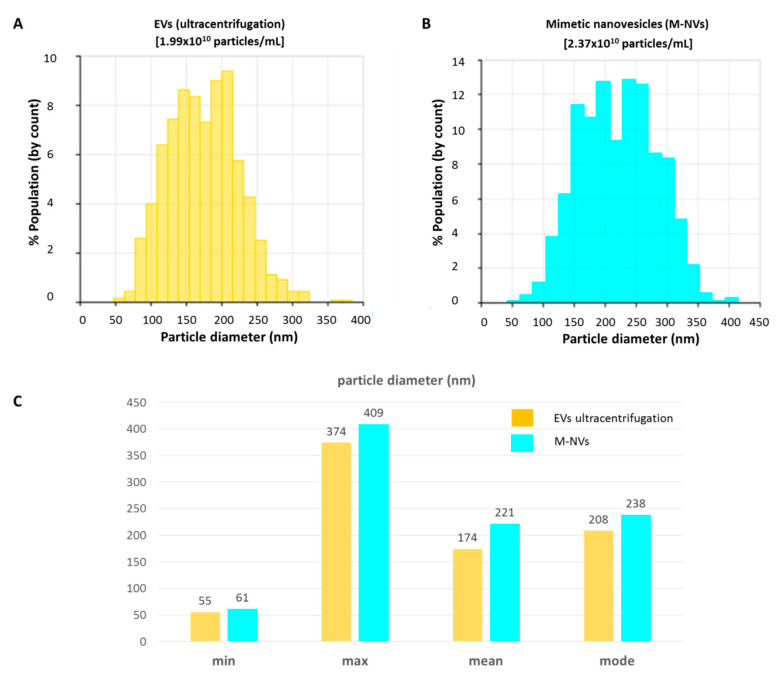
Analysis of T98G-derived EVs size and concentration. (**A**) EVs isolated by ultracentrifugation and (**B**) by plasma membranes reassembling protocol (M-NVs) were analysed through TRPS (qNano Gold, Izon). (**C**) Histogram of size distribution values of minimum, maximum, mean and mode values of the analysed nanoparticles (*n* = 2000).

**Figure 2 cells-09-01626-f002:**
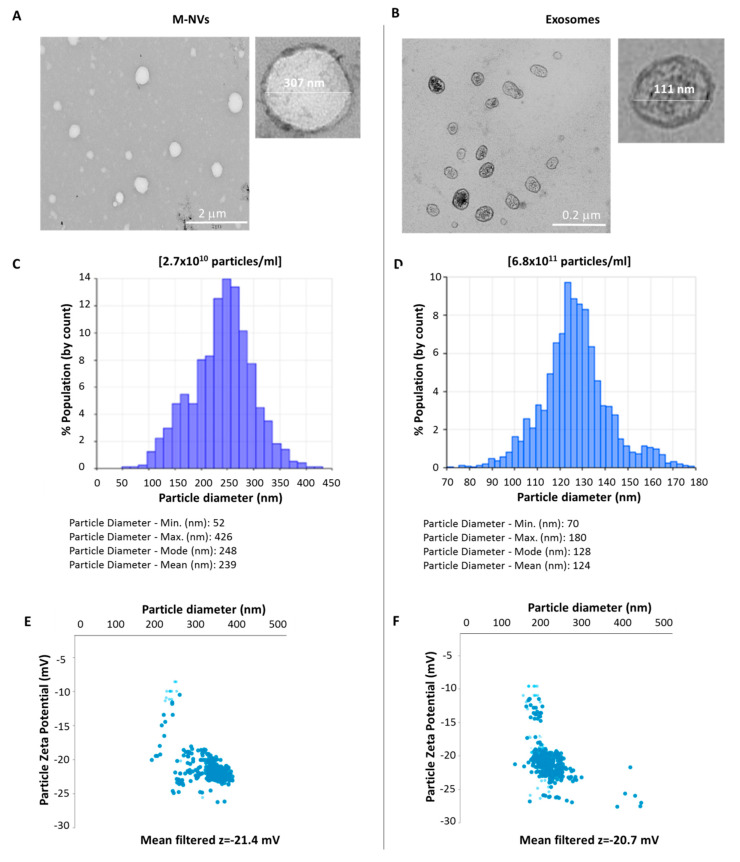
TEM ultrastructural analysis (**A**,**B**), size and concentration (**C**,**D**) and z-charge (**E**,**F**) comparison between membranes reassembling mimetic nanovesicles (M-NVs) and exosomes both isolated form T98G cells, determined through TRPS analysis (qNano Gold, Izon).

**Figure 3 cells-09-01626-f003:**
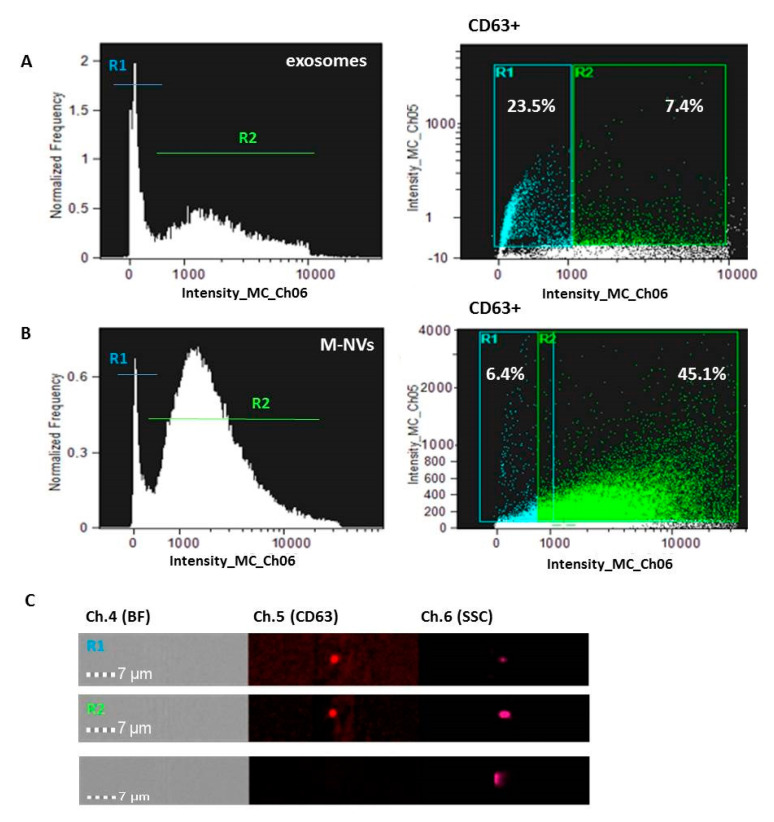
CD63 expression in R1 and R2 sub-populations from T98G exosomes (**A**) and M-NVs (**B**) by Amnis ImageStreamX MarkII flow cytometer. Examples of CD63 + and CD63- (Ch.5) EVs (60×) are reported (**C**). BF = Brightfield; SSC = scatterplot. A total of 25,000 particles were acquired.

**Figure 4 cells-09-01626-f004:**
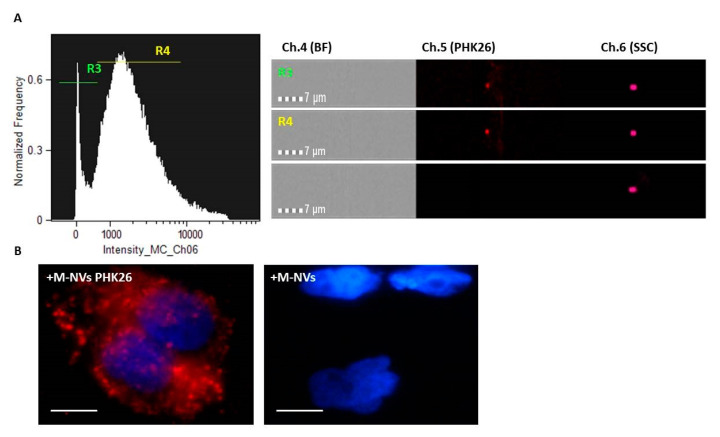
Amnis ImageStreamX MarkII flow cytometry and fluorescence microscope analysis of M-NVs stained with the red fluorescent PHK26 dye (**A**). PHK26 decorated M-NVs and unstained M-NVs were then separately administered to growing T98G cells. Cells were fixed after 24 h, DAPI stained and visualized under Nikon Eclipse TS100 fluorescence microscope (**B**) (100×, scale bars = 10 µm). For cytometry, a total of 25,000 particles were acquired.

**Figure 5 cells-09-01626-f005:**
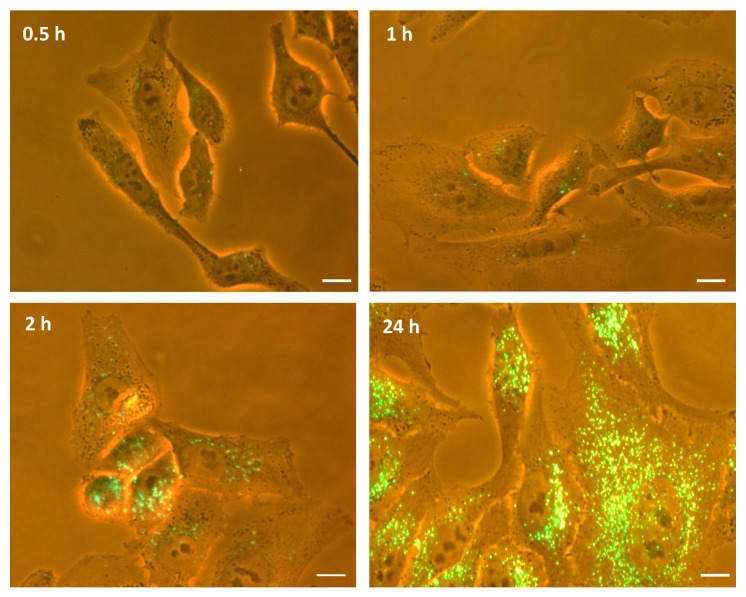
Kinetics of internalization of M-NVs stained with PHK67 dye into T98G cells at different time intervals (i.e., 0.5–1–2–24 h) visualized by inverted fluorescence microscope (40× magnification). Scale bar = 10 µm.

**Figure 6 cells-09-01626-f006:**
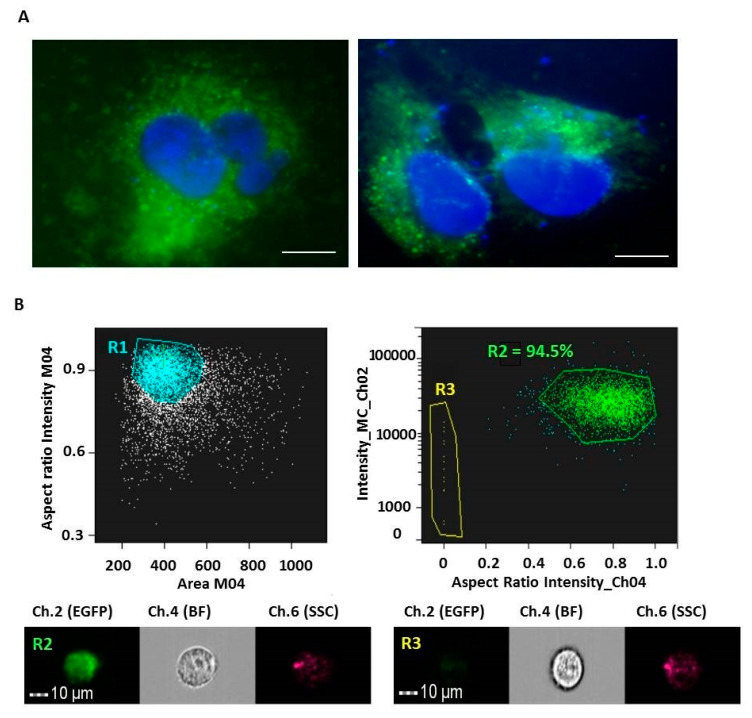
Fluorescence microscope and flow cytometric examinations of T98G cells after administration of electroporated membranes-generated vesicles (M-NVs, ~2 × 10^7^ particles) with pEGFP vector (0.5 µg). Cells (~2 × 10^3^) were fixed after 72 h p.t, DAPI stained and visualized by fluorescence microscope (100× magnification, scale bars = 10 µm) (**A**). Amnis ImageStreamX MarkII flow cytometry analysis was used to quantify the uptake efficiency (as percentage of fluorescent cells) of pEGFP electroporated M-NVs (~2 × 10^7^ particles) into T98G cells (~5 × 10^6^). As a result, R1 single cells subpopulation (*n* = 5000) showed a 94.5% of green fluorescent signals (**R2**). **R2** and **R3** representative cells were visualized using 40× magnification (**B**).

**Figure 7 cells-09-01626-f007:**
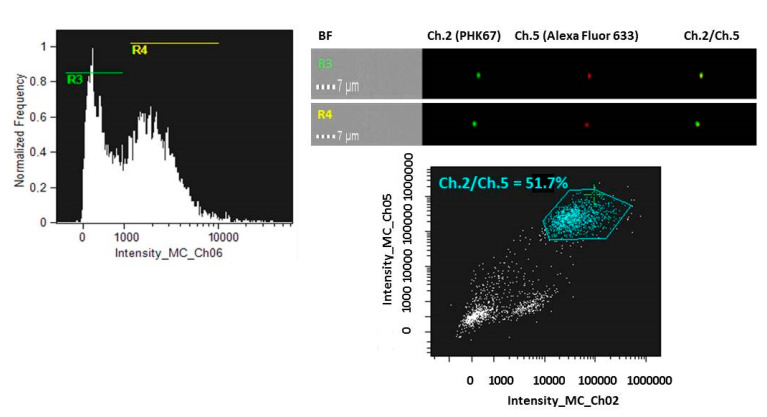
Amnis ImageStreamX MarkII flow cytometry analysis of membranes generated vesicles (M-NVs) stained with green fluorescent PHK67 dye and encapsulated with an Alexa Fluor 633 conjugated secondary antibody, visualized in Brightfield (**BF**), Ch.2 and Ch.5 fluorescent channels, respectively at 60× magnification. For cytometry, a total of 15,000 particles were acquired.

**Figure 8 cells-09-01626-f008:**
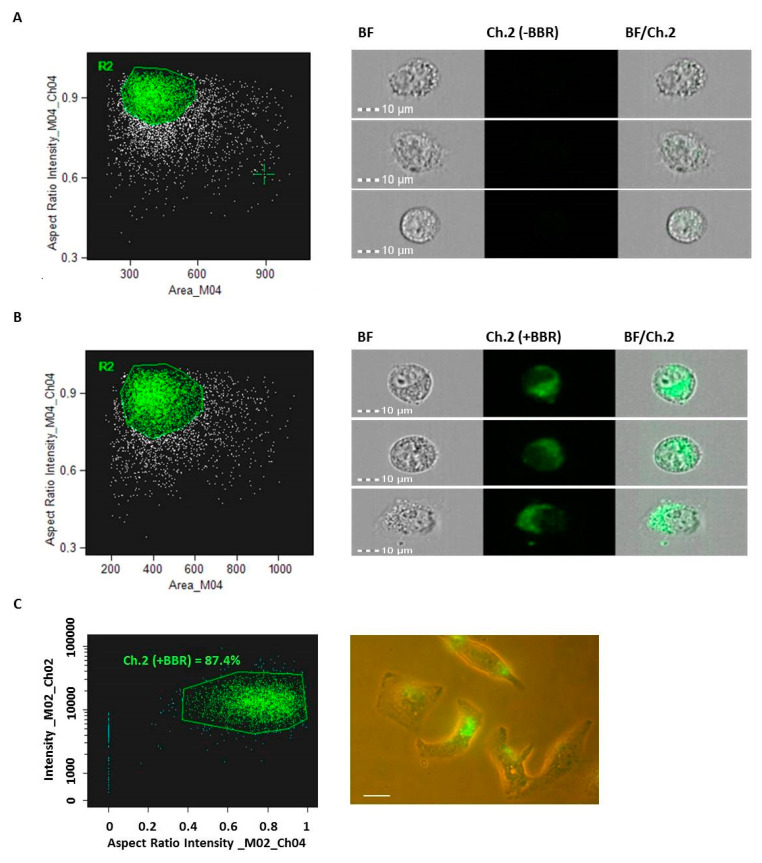
Amnis ImageStreamX MarkII flow cytometry analysis of T98× cells (R2 gates = 5000 single cells each) after administration of plasma membranes generated nanovesicles (M-NVs) (**A**) or with Berberine (BBR) encapsulated M-NVs, visualized in Brightfield (BF) and Ch.2 fluorescent channels (40×) (**B**). The percentage of BBR-positive M-NVs were evaluated by flow cytometry (i.e., 87.4% of R2 gated cells, C left panel); these M-NVs were then administered overnight to growing T98G cells and observed using Nikon Eclipse TS100 inverted fluorescent microscope (**C**, right panel) (40×, scale bar = 10 µm).

**Figure 9 cells-09-01626-f009:**
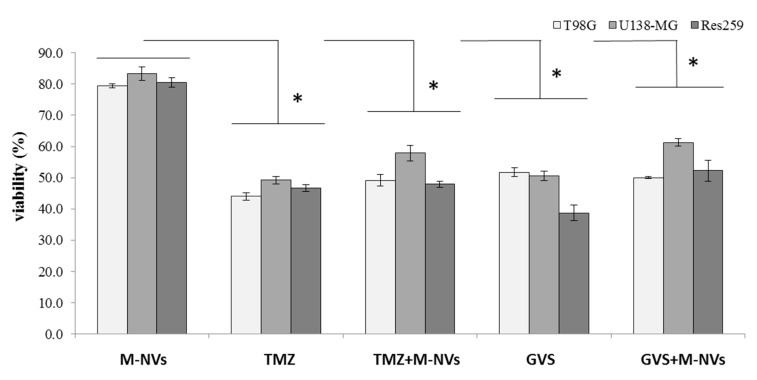
Cytofluorimetric viability assays in three different human astrocytoma cell lines (i.e., T98G, U138-MG and Res259) at 24 h after direct administration of mock plasma membranes generated vesicles (MVs), Temozolomide (TMZ) or Givinostat (GVS) and M-NVs loaded with TMZ or GVS separately (each 50 µM). Experiments were in triplicates analyzing 5000 cells for each treatment and asterisks indicate *p* < 0.05, Anova One-way compared to mock samples.

**Figure 10 cells-09-01626-f010:**
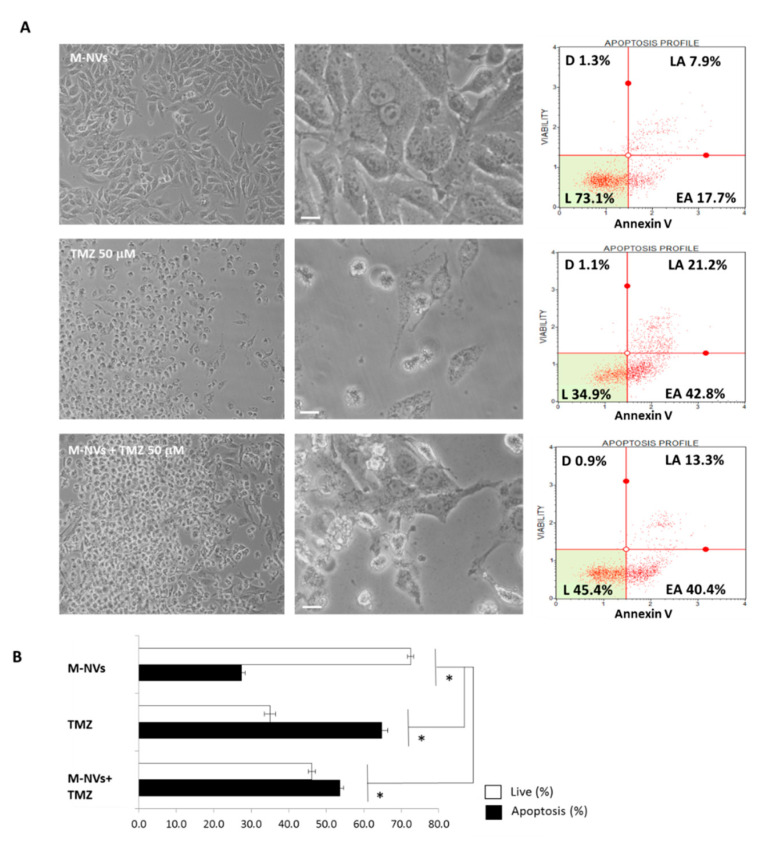
Flow cytometry analysis of apoptotic induction. (**A**) Annexin V expression in T98G cells after 24-h administration of mock membranes-derived vesicles (M-NVs), Temozolomide (TMZ 50 µM) and M-NVs encapsulated with equal TMZ concentration (L = live; D = dead; EA = early apoptotic; LA = late apoptotic cells percentages). Microscope contrast phase images (magnifications, 10× and 40×, scale bars = 10 µm). (**B**) Summary histogram of apoptosis *vs*. live percentages according to Annexin V assay. Experiments were in triplicates analyzing 5000 cells for each treatment and asterisks indicate *p* < 0.05 Anova One-way compared to mock M-NVs samples.

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
