# Peer review of "Development of Artificial Plasma Membranes Derived Nanovesicles Suitable for Drugs Encapsulation"

_cells, 2020, doi:10.3390/cells9071626_

Round 1

Reviewer 1 Report

Manuscript entitled “Development of artificial plasma membranes derived nanovesicles suitable for drugs encapsulation” describes the preparation of mimetic biologically functional nanovesicles. The topic is interesting, since it presents the alternative way of drug delivery into tumour cells, though the in vivo proof of concept is missing. However, the presented results are clear and sound. I have only few questions to answer and clarify.

In methods, there should be stated, how many batches of M-NVs were prepared and whether all experiments were done with one batch or the results are from several experiments and not only technical replicates.

For uptake of M-NVs some information about the efficiency of labelling (% of cells taking up the vesicles) will be beneficial.

Did you see any cytotoxic effect of Berberine on your cells? It should have also cytotoxic effect on tumour cells.

Do you know if the drug delivery via M-NVs is selective for tumour cells? I.e. whether the same level of apoptosis and decrease in cell viability will be observed in non-tumorous cells.

Minor. Figure S2 has very weakly visible, better resolution or thicker lines is recommended.

Author Response

In methods, there should be stated, how many batches of M-NVs were prepared and whether all experiments were done with one batch or the results are from several experiments and not only technical replicates.

This has been clarified in the revised version of the Methods. In summary, we started from a single cell sub-confluenting  plate, splitting 50% into identical plates and collecting generated M-NVs from the former, and exosomes or EVs by ultracentrifugation from the latter. Before any EVs isolation, in each plate cells were counted by Muse Cytofluorimeter as reported in the text to uniform EVs isolation procedures yields. As additional experiments, not reported in this contribution, we have positively assayed weekly frozen RIPA-lysed cells membranes fragments in their capability to generate functional M-NVs. Therefore, for convenience, different batches of precursor M-NVs can be prepared and next incubated with drugs molecules.

For uptake of M-NVs some information about the efficiency of labelling (% of cells taking up the vesicles) will be beneficial.

This important issue has been underlined in particular in Figure 6, where T98G cells (5x106) were incubated with M-NVs (2x107 particles) previously electroporated with pEGFP vector (0.5 microg).

According with Amnis flow cytrofluorimetric evaluation, 94.5% of  focused cells (n=5,000, R2/R1 gated) displayed scorable green fluorescent signals. Coherent semi-quantitative results were also obtaining through fluorescent microscope examinations, as illustrated in Figure 5 right-lower panel showing that nearly all visualized cells displayed fluorescent spots. Interestingly, these M-NVs were relatively stable in time duration, i.e. after 72 hours post-treatment.

Did you see any cytotoxic effect of Berberine on your cells? It should have also cytotoxic effect on tumour cells.

Berberine chloride showed in the reported astrocytoma established cell lines cytotoxic effect with IC50 at about 50 microM, after typically 48 hours of direct cell culture incubation.  However, in our study, using mimetic EVs as drug carrier, we preferred to assay lower doses, i.e. a final concentration of 10 microM. Preliminary results are suggesting a more effective cytotoxic effect in high grade glioma cells (i.e. WHO IV) compared to pilocytic astrocytoma-derived cells (Res186) and low grade astrocytoma (Res259). Lesser cytotoxic effect were assayed, as preliminary ongoing experiments, in normal human primary fibroblast cells.

Do you know if the drug delivery via M-NVs is selective for tumour cells? I.e. whether the same level of apoptosis and decrease in cell viability will be observed in non-tumorous cells.

This is a very crucial and important issue. After the preliminary feasibility demonstration, object of the present manuscript, we are planning several experiments comparing normal astrocytes vs tumor astrocytic cells. Preliminary and confidential results, comparing rat normal astrocytes and glioblastoma established cells (i.e. T98G and U87-MG) are suggesting a better uptake efficiency of drugs loaded M-NVs into tumor cells. These related results will be the topic of a next contribution.

Minor. Figure S2 has very weakly visible, better resolution or thicker lines is recommended.

As rightly suggested, Figure S2 has been modified accordingly and attached as PDF file

Reviewer 2 Report

The authors described a method to generate biological functional mimetic nanovesicles. It is an interesting work. But I have some questions about the authors’ work.
1. In this method, how to control the size of M-NV, or which factor can affect the size of M-NV? In this manuscript, only one size has been reported by authors. Does it mean this is the only size that can be obtained with this method? If not, I suggest authors do more work about the size control.
2. What is the drug loading efficiency in the Berberine, Temozolomide, and Givinostat treatment? Did the encapsulation of drugs affect the size or zeta potential of M-NV?

Author Response

  1. In this method, how to control the size of M-NV, or which factor can affect the size of M-NV? In this manuscript, only one size has been reported by authors. Does it mean this is the only size that can be obtained with this method? If not, I suggest authors do more work about the size control.

As documented in Figure2 (panel C) the size range of the mimetic generated EVs was 50-450 nm. We adopted this range interval to have a sufficiently clear size difference from the known exosome subclass of EVs, whose range is typically from 30-140 nm. The wider size range adopted was also selected with the purpose to generate larger cargo vesicles with the potential to host drugs molecules within, as suggested by TEM evidences reported in Figure 2A. According to our initial experiments in the setting up of the method, after cells lysis and the spontaneous reassembly of nanovescicles, one method to eventually select different size ranges of EVs is to purify the suspension using filters with different pores. For example, M-NVs passed through a 0,22 μM filter produced a vesicles population with similar diameters of exosomes. Differently, in the submitted manuscript size selection was based using a of 0.45 μm pore filter apparatus. This method of generation of mimetic nanovesicles is therefore easily adaptable to select the desired size range of EVs.

  1. What is the drug loading efficiency in the Berberine, Temozolomide, and Givinostat treatment? Did the encapsulation of drugs affect the size or zeta potential of M-NV?

As reported in Figure S2, that has been also improved according to Referee1, we were able to calculate the efficiency of Berberine uptake, using its autofluorescence feature through fluorimetric analysis. According to the interpolation curve, the efficiency of Berberine incorporation into M-NVs was about 10% (comparing the Log F value of BBR 1 mM with the identical value of 2.77 obtained using a 10 fold amount of BBR). In addition, we calculated that each single M-NVs hosted about  2x10exp6 molecules in average.

This manuscript is a resubmission of an earlier submission. The following is a list of the peer review reports and author responses from that submission.